# Persistent Atrial Fibrillation in Elderly Patients: Limited Efficacy of Pulmonary Vein Isolation

**DOI:** 10.3390/jcm11206070

**Published:** 2022-10-14

**Authors:** Andreas A. Boehmer, Moritz Rothe, Celine Zezyk, Christina M. Soether, Bianca C. Dobre, Bernhard M. Kaess, Joachim R. Ehrlich

**Affiliations:** Department of Cardiology, St. Josefs-Hospital Wiesbaden, Beethovenstraße 20, 65189 Wiesbaden, Germany

**Keywords:** age, atrial fibrillation, cryoballoon, pulmonary vein isolation

## Abstract

(1) Background: Cryoballoon pulmonary vein isolation (cryoPVI) is established for symptomatic paroxysmal atrial fibrillation (AF) treatment, but its value in persistent AF is less clear. In particular, limited data are available on its efficacy in elderly patients (≥75 years) with persistent AF. Age is an important modifier of AF progression and represents a risk-factor for AF recurrence. (2) Methods: Prospective, single-center observational study to evaluate the impact of age on efficacy and safety of cryoPVI in elderly patients. Primary efficacy endpoint was symptomatic AF recurrence after 90-day blanking period. Primary safety endpoints were death from any cause, procedure-associated complications or stroke/transient ischemic attack. Median follow-up was 17 months (range 3–24). (3) Results: We included 268 patients with persistent AF (94 ≥ 75 years of age). Multivariate Cox regression analysis identified age as the only independent factor influencing AF recurrence in the overall cohort (*p* = 0.006). To minimize confounding bias in efficacy and safety analysis of cryoPVI, we matched younger and elderly patients with respect to baseline characteristics. At 24 months, primary efficacy endpoint occurred in 13/69 patients <75 years and 31/69 patients ≥75 years of age (24 months Kaplan–Meier event-rate estimates, HR 0.34; 95% CI, 0.19 to 0.62; log-rank *p* = 0.0004). No differences were observed in the occurrence of safety end points. (4) Conclusions: Elderly (≥75 years) patients with persistent AF undergoing cryoPVI had an approximately threefold higher risk of symptomatic AF recurrence than matched younger patients. Accordingly, other treatment modalities may be evaluated in this population.

## 1. Introduction

Atrial fibrillation (AF) is the most common cardiac arrhythmia and associated with increased morbidity, mortality and impaired quality of life [1,2]. Pulmonary vein isolation (PVI) is a well-established treatment of symptomatic AF and has demonstrated its superiority over anti-arrhythmic drug (AAD) therapy for sinus rhythm maintenance [3,4]. Cryoballoon-PVI (cryoPVI) is established for treatment of paroxysmal AF [5] and recent studies indicate comparable safety and efficacy of cryoPVI compared with radiofrequency (RF) PVI in patients with persistent AF [6,7].

In the context of current demographic development with a steadily ageing population, studies in elderly patients (≥75 years) with paroxysmal AF suggest that cryoPVI offers a similarly favorable outcome as in younger patients [8,9]. However, age is an important modifier of disease and may represent a “risk-factor” for AF recurrence. Beyond fibrosis, changes in cellular atrial electrophysiology occur with age and might translate into impaired clinical outcome [10,11].

In previous studies among elderly patients, little distinction was made between paroxysmal and persistent AF when analyzing predictors of AF recurrence after PVI. In particular, reliable data for the influence of age in patients with persistent AF are scarce. Therefore, the present study was designed in order to investigate the influence of age on the efficacy and safety of cryoPVI in patients with symptomatic persistent AF.

## 2. Methods

### 2.1. Study Design

The present trial was a prospective observational, single-center study with predefined primary efficacy and safety endpoints conducted at St. Josefs Hospital (Wiesbaden, Germany) over a period of 3 years (January 2018 to December 2021). We enrolled consecutive patients affected by symptomatic persistent AF undergoing cryoPVI. Initial PVI in this cohort was exclusively performed as cryoPVI. All patients provided written informed consent for the procedure and for study participation. The study was approved by the regional ethics committee (represented by the Landesaerztekammer Hessen, ethics vote 2019-1474-evBO) and follows the rules of the Declaration of Helsinki. Patients or the public were not involved in the design, or conduct, or reporting, or dissemination plans of our research.

#### Eligibility Criteria and Matching Criteria

Consenting patients over 18 years of age with symptomatic persistent AF and indication for a first-time PVI were eligible for inclusion. Pregnant patients or patients unable to consent were not treated.

To make the groups directly comparable with respect to potential confounders, each patient over 75 years of age was individually matched with a patient under 75 years of age at a 1:1 ratio for baseline characteristics that were statistically differentially distributed.

### 2.2. Endpoints

The primary efficacy endpoint was first recurrence of documented, symptomatic AF after a 90-day blanking period post-cryoPVI. The primary safety endpoint consisted of death from any cause, procedure-associated complications (procedure-related death, major groin site complications, pericardial effusion, cerebrovascular or systemic embolism, phrenic nerve palsy, nonfatal or fatal stroke/transient ischemic attack).

### 2.3. Procedure

CryoPVI procedure was performed in a standardized manner by an experienced physician (BMK or JRE). In brief, a 15 F (outer diameter) steerable sheath (FlexCath Advance, Medtronic^TM^, Dublin, Ireland) was introduced into the left atrium after a single trans-septal puncture (TSP). TSP was performed under fluoroscopic guidance, with 5000 units unfractionated heparin administered prior to TSP. An activated clotting time of 300–350 s (sec) was chosen as the target range. Non-vitamine K oral antagonists were given “minimally” interrupted (pause on the morning of cryoPVI).

Then, the second-generation Arctic Front (Medtronic^TM^) balloon was introduced, inflated in left atrium and advanced to the ostium of each pulmonary vein. The 28 mm balloon was used in all cases. Pulmonary vein signals were recorded with an octapolar, circular mapping catheter (Achieve, Medtronic^TM^). Occlusion of each vein was assessed with venous angiography. Right-sided veins were ablated during phrenic nerve stimulation with a decapolar catheter (Dynamic, Boston Scientific^TM^, Marlborough, MA, USA). Ablation was performed with a freeze duration of 180 s per pulmonary vein. When visualized online, time to isolation +120 s was used as dosing protocol. In veins without reliable signals, one 180 s freeze was applied. All veins were re-checked at the end of the procedure. Discontinuation of class I and III AAD at the end of the 90-day blanking period was encouraged and oral anticoagulation was performed according to the CHA_2_DS_2_-VASc score as per guidelines.

### 2.4. Follow-Up

Follow-up was aimed at detecting symptomatic relapses and was based on predefined trans-telephonic interviews at 3, 6, 12, 18 and 24 months, including visits to clinic or outpatient holter-ECGs when indicated by symptoms. Hospitalizations were recorded. Patients with the sensation of palpitations were scheduled for ECG registration.

### 2.5. Statistical Analysis

The Kaplan–Meier method was used to calculate event-rate estimates. Testing was performed using log-rank test and multivariate cox-regression analysis. Normally distributed variables were compared using a two-sample t-test. Dichotomous variables were compared using Fisher’s exact test. Significance level was set at 0.05 (5%). Mean values are presented with standard deviation. Statistical analyses were performed using PRISM software version 9 (GraphPad^TM^, San Diego, CA, USA) and SPSS version 28 (IBM^TM^, Armonk, NY, USA).

## 3. Results

### 3.1. Patient Population

A total of 268 consecutive patients with symptomatic persistent AF were enrolled and underwent cryoPVI. Of these, 94 patients were elderly according to our definition (≥75 years). Median follow-up time was 17 months (range: 3–24 months). Baseline characteristics differed in terms of higher proportion of women (*p* < 0.001), coronary artery disease (*p* < 0.001) and impaired renal function (*p* < 0.018) as well as lower proportion of obesity (BMI ≥ 30 kg/m^2^) (*p* < 0.001) in patients ≥75 years of age (Table 1). There was a statistical trend for higher proportion of hypertension (*p* = 0.068) in elderly patients. The proportion of patients still on AAD during and after blanking phase was comparable between both age groups.

### 3.2. Association of Clinical Parameters with Primary Efficacy Endpoint

By means of multivariate Cox regression analysis, the only characteristic independently associated with the occurrence of primary efficacy endpoint for the overall cohort was age (*p* = 0.006, Table 2).

### 3.3. Influence of Age on Primary Efficacy Endpoint

In view of significant age-related differences with regard to pre-existing cardiovascular conditions, we performed matching in order to establish maximum comparability. Matching was performed for the statistically different distribution of the variables gender, presence of obesity, coronary artery disease, impaired renal function and hypertension. Thus, 69 precise matches could be formed, which we used for further analysis (Table 3).

At 24 months, the primary efficacy endpoint occurred in 13/69 younger patients and 31/69 matched patients ≥75 years of age (24 months Kaplan–Meier event-rate estimates, hazard ratio [HR] 0.34; 95% confidence interval [CI], 0.19 to 0.62; log-rank *p* = 0.0004, Figure 1).

### 3.4. Primary Safety Endpoint

Safety endpoints were rare. One endpoint occurred in a patient in the young group (air embolism into right coronary artery with spontaneous resolution) and in none of the elderly patients. No deaths or strokes were observed in either group.

### 3.5. Procedural Parameters

Total procedure time (58 ± 19 min vs. 54 ± 15 min, *p* = 0.17), left-atrial dwell time (39 ± 15 min vs. 35 ± 10 min, *p* = 0.11), fluoroscopy time (7 ± 4 min vs. 7 ± 3 min, *p* = 0.8), contrast dye use (52 ± 36 mL vs. 45 ± 35 mL, *p* = 0.28) and radiation dose (285 ± 308 cGy*cm^2^ vs. 260 ± 160 cGy*cm^2^, *p* = 0.58) were similar between procedures performed for younger and elderly patients (Figure 2).

## 4. Discussion

This observational study prospectively compared efficacy and safety of cryoPVI in elderly patients (≥75 years). Our main findings were twofold: firstly, elderly patients with persistent AF showed an approximately threefold higher rate of symptomatic AF recurrence after cryoPVI compared with younger patients with persistent AF; secondly, the procedure was as safe in elderly patients as in younger patients.

### 4.1. PVI in Elderly Patients

In recent years, a number of studies have addressed PVI in elderly patients and reported widely varying success rates with different treatment regimens [8,9,12]. In particular, case numbers were small (<100) for cryoPVI and 1-year success rates varied from 30% to 87% [8,9,13]. With respect to treatment modality, there is evidence of comparable efficacy with shorter procedure duration of cryoPVI compared with radiofrequency PVI [14].

### 4.2. CryoPVI for Persistent AF

PVI has been the cornerstone of ablation procedures in patients with persistent AF since publication of the STAR-AF II trial [15]. In STAR-AF II, patients with persistent AF undergoing RF-PVI without additional lesion sets had a 12-month AF-free rate of 59% including patients on AAD. This rate decreased to 49% of patients without AAD after 1 year. This rate was comparable to patients undergoing additional ablation (lines or defragmentation).

The underlying pathophysiological basis among patients with persistent AF is heterogeneous. Some may have PV-dependent AF and can be treated with PVI while a significant proportion will need additional treatment. The duration of AF persistence over an arbitrarily chosen period (indicating paroxysmal vs. persistent AF as per guideline) may, in some cases, fail to identify patients that can successfully be treated with PVI alone.

The CHASE-AF study provided interesting insight into the prevalence of PV-dependent AF [16,17]. In roughly one-quarter of procedures the arrhythmia terminated during PVI and these patients with PV-dependent persistent AF were excluded from randomization. In CHASE-AF, freedom from AT after one ablation was ~64% in the PVI group with a substantial percentage of patients treated with additional AAD (~29%) at 12 months [16]. The proportion of patients on AAD in CHASE-AF compares unfavorably to our patient population where only ~6% of patients with persistent AF were on AAD after the blanking period. However, in our study patients who converted to sinus rhythm during or after PVI were not excluded from analysis and our study may have comprised patients with persistent but still PV-dependent AF.

Among other studies reporting the use of cryoPVI in persistent AF that did similarly not exclude patients converting to sinus rhythm during the procedure an off-drug freedom from AT ranging from 56% to 82% after 1 year have been reported [6,18,19]. This indicates that a large proportion of patients may in fact have PV-dependent persistent AF, which will benefit from PVI only. Clinical factors that help to identify patients likely to benefit from isolated PVI still need to be identified.

### 4.3. Pathophysiological Role of Age in Persistent AF

Epidemiological studies illustrate age as one of the main factors in the progression of the disease from paroxysmal to persistent and permanent AF [20].

Only limited mechanistic data exist regarding the influence of ageing on human atrial electrophysiology and its role for AF-related remodeling. Clinical electrophysiological studies in healthy atria showed an inverse correlation of age with wavefront propagation velocity measured by electro-anatomical mapping [21]. Ageing was associated with regional conduction slowing in another study [22] and studies accessing the role of age on atrial fibrosis have found a significant correlation between age and fibrosis burden in human atria [23,24]. Age-related increases in atrial fibrosis, therefore, may provide a possible explanation for the altered conduction velocities in the atria of elderly patients [11].

Only very sparse human basic data on functional electrophysiology exist. Studies in animals found decreased calcium current function and augmented potassium currents in ageing dogs [25], and indirect evidence for these changes as potential contributors (besides increased fibrosis) to AF vulnerability in old canines [26]. Similar changes have been observed in human atria showing a reduction in calcium channel subunits in elderly humans [11]. Data indicate a correlation of atrial tissue fibrosis estimation by delayed enhancement magnetic resonance imaging with ablation outcome [27]. Additionally, fibrosis may either occur as a consequence of atrial cardiomyopathy [28] or increasing age [23].

### 4.4. Alternative Therapeutic Strategies for Elderly Patients with Symptomatic Persistent AF

Against the background of epidemiological developments in ageing societies, the development of strategies to treat elderly patients with persistent AF is needed. Given the high rate of AF recurrence after PVI among elderly patients with persistent AF, alternative therapeutic options need to be sought. More extensive left atrial ablation such as defragmentation of complex fractionated atrial electrograms or box isolation of fibrotic areas might represent a suitable strategy [29]. However, in view of patient age, pacemaker implantation and His-bundle ablation may also represent a reasonable treatment option [30] that similarly needs to be tested.

### 4.5. Safety

No procedure-related death, cerebral ischemic events, major groin site-complication or cardiac tamponade were observed irrespective of age. In the present trial, total procedure time, left-atrial dwell time, fluoroscopy time, contrast dye and radiation dose for patients under and over 75 years of age were similar and significantly shorter than reported in other studies [5,6].

## 5. Strengths and limitations

To the best of our knowledge, this is the first study directly comparing cryoPVI in elderly versus younger patients with persistent AF. Strengths of our study include a highly homogeneous treatment approach and a homogenous study sample, with all cryoPVI procedures performed in a single center by two experienced electrophysiologists.

Some limitations should be mentioned. First, our study is purely observational und causal inferences cannot be drawn. Second, clinical follow-up was based on trans-telephonic interviews and triggered ECGs if symptoms suggestive for AF-recurrence occurred. Thus, asymptomatic AF-recurrences were not documented, and the total success rate is certainly overestimated from a rhythmological point of view. However, this is the case for both older and younger patients and thus should not introduce a systematic bias into the comparison of groups. Moreover, from an elderly patient’s perspective a symptomatic relapse is a much more meaningful event than an asymptomatic recurrence. Fourth, we did not record left atrial or PV voltage information, precluding mechanistic insights, in particular the evaluation of atrial scarring or slow conduction. Last, this trial was not controlled for AAD use.

## 6. Summary

Elderly patients with persistent AF had an approximately threefold higher risk of symptomatic AF relapse after cryoPVI than matched patients of younger age. Further randomized studies are necessary to identify optimal therapeutic approaches for treatment in this demographically important population.

## Figures and Tables

**Figure 1 jcm-11-06070-f001:**
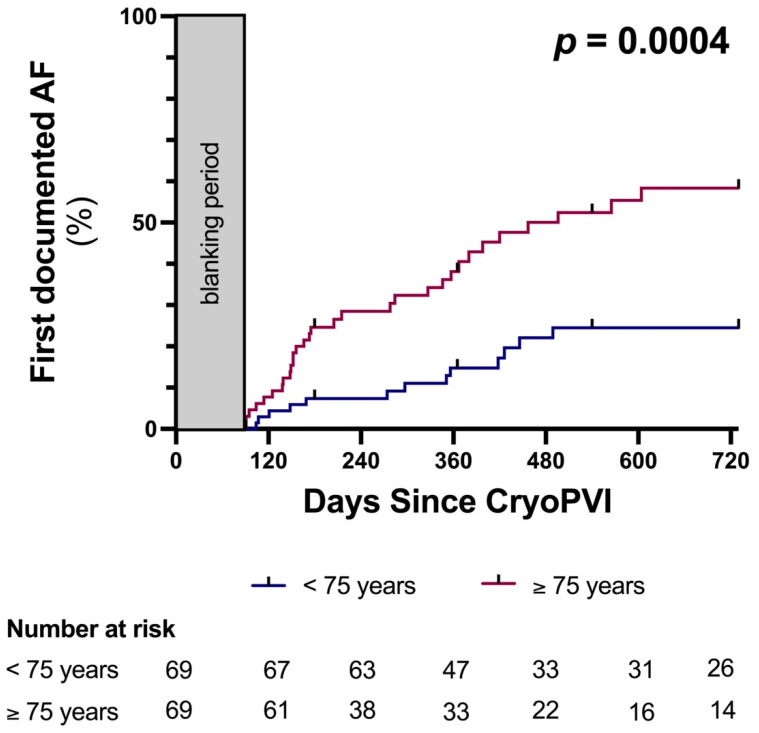
Kaplan-Maier estimates for patients aged over (purple) and under (blue) 75 years of age. AF, atrial fibrillation; cryoPVI, cryoballoon pulmonary vein isolation.

**Figure 2 jcm-11-06070-f002:**
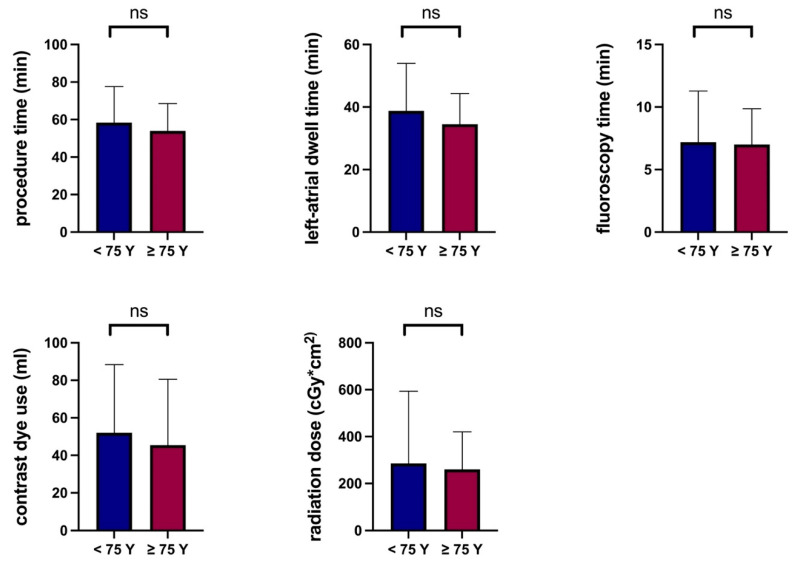
Comparison of procedural parameters between patients <75 and ≥75 years of age. Y, years; ns, not significant.

**Table 1 jcm-11-06070-t001:** Baseline characteristics according to age.

Characteristic	Age < 75 years (*n* = 174)	Age ≥ 75 years (*n* = 94)	*p*
Age (years)	64.5 ± 8	79.6 ± 3	<0.001
Male sex	118 (67.8%)	44 (46.8%)	0.001
Body mass index ≥ 30 kg/m^2^	57 (32.8%)	12 (12.8%)	<0.001
Antiarrhythmic drugs			
(during/after blanking period)			
overall	82 (47.1%)/17 (9.8%)	43 (45.7%)–6 (6.4%)	0.9/0.5
Flecainide	29 (16.7%)/8 (4.6%)	13 (13.8%)–0 (0%)	0.6/0.055
Amiodarone	53 (30.5%)/9 (5.2%)	28 (29.8%)–6 (6.4%)	0.99/0.78
Dronedarone	2 (1.2%)/0 (0%)	0 (0%)–0 (0%)	0.54/0.99
CHA_2_DS_2_-VASc score	2.3 ± 1.3	3.9 ± 1.1	<0.001
LA-Diameter (mm)	43 ± 9	45 ± 11	0.24
Coronary artery disease	23 (13.2%)	30 (31.9%)	<0.001
Heart failure (LVEF ≤ 40%)	36 (20.7%)	22 (23.4%)	0.64
Hypertension	116 (61.3%)	73 (78.8%)	0.068
Hyperlipidemia	25 (14.4%)	13 (13.8%)	0.99
Diabetes	21 (12.1%)	12 (12.8%)	0.85
Impaired renal function	23 (13.2%)	24 (25.5%)	0.018
Previous stroke	8 (4.6%)	8 (8.5%)	0.28

Continuous data are presented as mean value ± standard deviation, numbers and percentage (in brackets) are given for absolute values: 90-day blanking period, LVEF–left ventricular ejection fraction.

**Table 2 jcm-11-06070-t002:** Influence of clinical characteristics on primary efficacy endpoint (cox-regression model).

Characteristic	Persistent AF			
	*p*	Exp (B)	95% CI for Exp (B)	
Age (years)	0.006	1.04	1.01	1.07
Male sex	0.53	0.86	0.54	1.38
BMI > 30 km/m^2^	0.73	1.11	0.61	2.01
LA Diameter (mm)	0.33	0.99	0.97	1.01
Coronary artery disease	0.55	0.82	0.43	1.55
Heart failure (LVEF ≤ 40%)	0.94	1.03	0.56	1.90
Hypertension	0.07	0.65	0.40	1.04
Hyperlipidemia	0.66	0.85	0.43	1.69
Diabetes	0.44	0.69	0.26	1.81
Impaired renal function	0.41	0.80	0.47	1.36

**Table 3 jcm-11-06070-t003:** Baseline characteristics and primary efficacy endpoint according to age after matching.

Characteristic	Age < 75 years (*n* = 69)	Age ≥ 75 years (*n* = 69)	*p*
Age (years)	65.7 ± 8.4	79.5 ± 3.1	<0.001
Primary efficacy endpoints at 24 months	13 (18.8%)	31 (44.9%)	<0.001
Male sex	37 (53.6%)	37 (53.6%)	0.99
Body mass index ≥ 30 kg/m^2^	5 (7.3%)	5 (7.3%)	0.99
Antiarrhythmic drugs			
(during/after blanking period)			
overall	30 (43.5%)/4 (5.8%)	27 (39.1%)–6 (8.7%)	0.73/0.75
Flecainide	9 (13%)/2 (2.9%)	9 (13%)–0 (0%)	0.99/0.5
Amiodarone	21 (30.4%)/2 (2.9%)	17 (24.7%)–6 (8.7%)	0.57/0.27
Dronedarone	0 (0%)/0 (0%)	1 (1.5%)–0 (0%)	0.99/0.99
CHA_2_DS_2_-VASc score	2.7 ± 1.4	3.8 ± 1.1	<0.001
LA-Diameter (mm)	44 ± 9	44 ± 9	0.86
Coronary artery disease	18 (26.1%)	18 (26.1%)	0.99
Heart failure (LVEF ≤ 40%)	14 (20.3%)	14 (20.3%)	0.99
Hypertension	53 (76.8%)	53 (76.8%)	0.99
Hyperlipidemia	8 (11.6%)	10 (14.5%)	0.80
Diabetes	6 (8.7%)	6 (8.7%)	0.99
Impaired renal function	15 (21.7%)	15 (21.7%)	0.99
Previous stroke	5 (7.3%)	8 (11.6%)	0.56

Continuous data are presented as mean value ± standard deviation, numbers and percentage (in brackets) are given for absolute values: 90-day blanking period, LVEF–left ventricular ejection fraction.

## Data Availability

The raw data presented in this study are available on request from the corresponding author. The data are not publicly available due to requirements of ethics guidelines.

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
