# Peer review of "Persistent Atrial Fibrillation in Elderly Patients: Limited Efficacy of Pulmonary Vein Isolation"

_jcm, 2022, doi:10.3390/jcm11206070_

Round 1

Reviewer 1 Report

Boehmer and colleagues present this matched analysis from a propective cohort of 268 patients undergoing cryoPVI alone for persistent AF, comparing 94 patients of age 75 or greater (elderly) with those younger than 75.  Despite matching for the factors listed in Table 3, the elderly cohort had a significantly higher rate of recurrent AF during the follow-up period.  The authors conclude that PVI alone may not be sufficient to fully address AF substrate in the elderly with persistent AF.

This is a well-done analysis and contributes to the literature regarding ablation of persistent AF.

Minor comment:

1. Table 1 lists a smaller proportion of BMI high in elderly group, but text (line 4 of Results first paragraph) implies more obesity in the elderly (just need to clear up the wording of the sentence).

Author Response

Reviewer: Table 1 lists a smaller proportion of BMI high in elderly group, but text (line 4 of Results first paragraph) implies more obesity in the elderly (just need to clear up the wording of the sentence)

Response to reviewer: The authors are grateful for the reviewer's attention to detail. We have adjusted the sentence as follows:

"Baseline characteristics differed in terms of higher proportion of women (P<0.001), coronary artery disease (P<0.001) and impaired renal function (P<0.018) as well as lower proportion of obesity with BMI ≥30 kg/m2(P<0.001) in patients ≥75 years of age (table 1)."

Reviewer 2 Report

It's generally well-written article.

I have a few comments.

#1. It would be better to incorporate the overall relapse rate (primary outcome) to Tables.

#2. In Table 2,

Confidence intervals must be included.

Some variables require units (eg. age (years), LA diameter (mm), etc.).

#3. Because the sample size is small and the number of primary outcomes is small, it may be statistically problematic to input many variables in multivariate analysis at once. How about excluding variables that are not significantly different at baseline analysis (eg. hyperlipidemia, etc.)?

Author Response

Reviewer: (1) It would be better to incorporate the overall relapse rate (primary outcome) to Tables.

Author response: Thank you for considering this aspect. We have adjusted the tables accordingly. 

Reviewer: (2) In Table 2, Confidence intervals must be included. Some variables require units (eg. age (years), LA diameter (mm), etc.).

Author response: The authors are grateful for the reviewers attention to detail. We have included 95% CI into table 2 and added missing variables.

Reviewer: (3) Because the sample size is small and the number of primary outcomes is small, it may be statistically problematic to input many variables in multivariate analysis at once. How about excluding variables that are not significantly different at baseline analysis (eg. hyperlipidemia, etc.)?

Author response: We thank the reviewer for this really valuable input. We are aware that, especially with a small number of cases, multivariate analyses/multiple testing can yield primarily false positive, but also false negative results. Therefore, within the statistical analysis for this manuscript, multivariate analyses with exclusion of "less important" parameters as well as univariate analyses were performed according to the reviewer's thought. Nevertheless, exactly the same picture emerged here as when all the listed parameters were taken into account: The influence of the variable "age" is so highly significant that it completely overweights all other parameters. It was therefore important to us to be able to present this in the form of this multivariate analysis.